# Background Point Filtering of Low-Channel Infrastructure-Based LiDAR Data Using a Slice-Based Projection Filtering Algorithm

**DOI:** 10.3390/s20113054

**Published:** 2020-05-28

**Authors:** Ciyun Lin, Hui Liu, Dayong Wu, Bowen Gong

**Affiliations:** 1Department of Traffic Information and Control Engineering, Jilin University, Changchun 130022, China; linciyun@jlu.edu.cn (C.L.); huiliu19@mails.jlu.edu.cn (H.L.); 2Department of Engineering Training Center, Changchun Institute of Technology, Changchun 130012, China; 3Texas A&M Transportation Institute, Texas A&M University, College Station, TX 77843, USA; J-Wu@tti.tamu.edu; 4Jilin Engineering Research Center for ITS, Changchun 130022, China

**Keywords:** background points filtering, infrastructure-based LiDAR, slice-based projection, 3-D point cloud

## Abstract

A light detection and ranging (LiDAR) sensor can obtain richer and more detailed traffic flow information than traditional traffic detectors, which could be valuable data input for various novel intelligent transportation applications. However, the point cloud generated by LiDAR scanning not only includes road user points but also other surrounding object points. It is necessary to remove the worthless points from the point cloud by using a suitable background filtering algorithm to accelerate the micro-level traffic data extraction. This paper presents a background point filtering algorithm using a slice-based projection filtering (SPF) method. First, a 3-D point cloud is projected to 2-D polar coordinates to reduce the point data dimensions and improve the processing efficiency. Then, the point cloud is classified into four categories in a slice unit: Valuable object points (VOPs), worthless object points (WOPs), abnormal ground points (AGPs), and normal ground points (NGPs). Based on the point cloud classification results, the traffic objects (pedestrians and vehicles) and their surrounding information can be easily identified from an individual frame of the point cloud. We proposed an artificial neuron network (ANN)-based model to improve the adaptability of the algorithm in dealing with the road gradient and LiDAR-employing inclination. The experimental results showed that the algorithm of this paper successfully extracted the valuable points, such as road users and curbstones. Compared to the random sample consensus (RANSAC) algorithm and 3-D density-statistic-filtering (3-D-DSF) algorithm, the proposed algorithm in this paper demonstrated better performance in terms of the run-time and background filtering accuracy.

## 1. Introduction

As a type of active vision sensor, light detection and ranging (LiDAR) is a 3-D point cloud imaging vision sensor with the advantages of insensitivity to external light changes, strong adaptability to complex environments, wide coverage, and is informative. Currently, the most typical application of LiDAR in intelligent transportation systems is to detect road and traffic information for automatic driving vehicles [1,2,3,4]. Despite the rapid development of technologies related to self-driving cars in recent years, we cannot realize automatic driving by merely using on-board detectors. Fatal accidents may occur if the detectors of an autonomous vehicle make errors. For example, Tesla and Uber have experienced fatal accidents during their self-driving tests [5,6]. Therefore, it is very important for roadside equipment to monitor road traffic in mixed connected and unconnected traffic conditions. Roadside equipment sends road information to road users to form a cooperative vehicle infrastructure system that can improve traffic efficiency and safety [7].

At present, many roadside detection devices collect traffic information from cameras [4,8,9]. The cameras perceive traffic flow and road surface information by obtaining the brightness in combination with red, green and blue (RGB). However, they cannot directly obtain the depth information. Some algorithms were developed to convert the image information into the corresponding depth information [10], which carried out the 3-D detection and trajectory tracking of road users. However, there are some accuracy risks, as the data provided by a camera can be easily affected by light, and the accuracy at night is relatively low. On the contrary, LiDAR can directly obtain the credible depth information without the influence of light and can accurately obtain the location of objects and their 3-D shape features. In this regard, more benefits can be obtained by deploying a roadside LiDAR sensor (also called an infrastructure-based LiDAR sensor).

The LiDAR sensor can obtain richer and more detailed information of traffic flow characteristics and parameters than traditional traffic detectors, such as video, infrared, radar, radio frequency, and geomagnetic sensors. It is possible to extract the high-resolution fine-grained position, direction, speed, and even the trajectory of each vehicle or pedestrian within the LiDAR scanning range. The micro-level traffic data detected by a roadside LiDAR sensor could be a valuable data input for a vehicle–infrastructure cooperation system, connected/autonomous vehicle systems, vehicle-to-pedestrian crash reduction analysis, smart traffic signals, and other various traffic application fields [11].

In addition, compared with the high-channel LiDAR that are currently deployed on autonomous vehicles or other traditional traffic flow sensors, the low-channel LiDAR has already proved its great potential in traffic information collection due to the low cost, high accuracy and reliability, high reusability, and portability [12,13]. For example, a magnetometer is one of the commonly used traffic flow sensors. The price of a magnetometer is about U.S. $400 per unit in the area of traffic engineering in China. Installing a magnetometer in each lane is required to detect traffic flow parameters. Generally, if considering two or three lanes in each direction of the road, four or six magnetometers are usually needed to realize the overall traffic flow detection on the road segment.

However, only one LiDAR sensor is needed to completely detect the traffic flow parameters on a road segment or at an intersection due to the 360-degree detection coverage. In addition, the price of low-channel LiDAR sensors is expected to drop significantly to about U.S.$100 if massive production is realized in the near future. Therefore, the overall cost of the solutions based on low-channel LiDAR sensors will be lower than most current traffic flow detectors. In this regard, the low-channel LiDAR sensor will have a broad application in the fields of traffic flow detection, early safety warnings of people or vehicles in the mixed environment of connected and unconnected traffic, and many others.

Real-time detecting and tracking of fast-moving vehicles is very important for transportation applications. For example, a fast algorithm can give a timely alarm when a vehicle and other objects are about to collide. For another example, if the algorithm can quickly perceive dangerous situations and release safety messages to other vehicles in time when the vehicle suddenly brakes or has fatal faults, it can reduce the occurrence of accidents as much as possible. The point cloud generated by LiDAR scanning is very informative, including buildings, trees, road users, and the ground. The amount of points used to describe road users in a point cloud is usually less than 1/10. The amount of points directly affects the computational speed and operational efficiency in identifying and tracking road users [14]. Therefore, it is necessary to remove worthless points from the point cloud by using a suitable background filtering algorithm.

Academic researchers and industry engineers are developing many background point filtering algorithms for on-board LiDAR data. Asma [15] used an octree-based occupancy grid representation to model the dynamic environment around the vehicle and to detect moving objects based on inconsistencies between scans. Alireza [16] built a voxel-grid model for static and moving obstacles’ detection by using discriminative analysis and ego-motion information. Dewan [17] presented a novel model-free approach for detecting and tracking dynamic objects in 3-D LiDAR scans obtained by a moving sensor. This relied only on motion cues and did not require any prior information regarding the objects. These methods mentioned above can all remove background points to obtain moving object information around the vehicle.

There is some valuable information for the vehicle in static objects, such as curbstones or other ground information. Zhang [18] developed a real-time curb detection method for self-driving based on the road segmentation. The point cloud data were first processed to distinguish on-road and off-road areas. Then, they proposed a sliding-beam method to segment the road by using the off-road data. The curb information formed the foundation of the decision-making and path planning for autonomous driving.

Multiple LiDARs outperform the single LiDAR in terms of object classification [19]. Therefore, Cao [20] proposed the occupancy-rate-based filtering method for filtering point clouds generated by fusing multiple LiDARs. Although numerous background filtering algorithms of on-board LiDAR have been developed, those algorithms cannot be applied to infrastructure-based LiDAR sensors directly as the infrastructure-based LiDAR and on-board LiDAR differ in the data quality, density, and expected application targets. In particular, on-board LiDAR are designed to detect the environment around the vehicle, which needs to work with video cameras, high-resolution 3-D maps, and high-precision global position system (GPS) to provide supportive data. Infrastructure-based LiDAR works individually to detect wild range traffic flow information, such as a whole intersection [17,18].

With a roadside LiDAR sensor, Wu [21] identified ground points by a channel-based filtering algorithm, which is a combination of raster-based clustering, distance-based/neighborhood-based filtering, and a slope-based method. The algorithm worked well for both flat roads and non-flat roads. Vosselman [22] differentiated point clouds into ground points and non-ground points according to the height difference between each. Konolige [23] used the random sample consensus (RANSAC) algorithm to select many points and selected a plane that was the most close to selected points as the ground plane.

Zhang [24] proposed the slope-based algorithm to distinguish ground points and non-ground points by the points’ differences in characteristics. Ground surfaces are relatively continuous and smooth while the shapes of non-ground features, such as trees and buildings, have a sharp change in the slope. However, those methods mentioned above only distinguish ground points and non-ground points. We still need to extract valuable objects (such as road users) from the point cloud. Wu [25] also introduced an automatic background filtering method named 3-D density-statistic-filtering (3-D-DSF). This method can filter both static background and vibrating backgrounds efficiently by the threshold of cube-point-density (TD) for learning automatically.

For the 3-D-DSF method, all static background points were removed from the data clouds, including curbstones, tree trunks, and other fixed background objects. As a result, some valuable points for traffic were also filtered out of the data set. Furthermore, these algorithms were based on multi-frame point cloud data, which are not suitable for on-line traffic object identification and trajectory tracking. In addition, there are other filtering algorithms based on high-density point clouds [26,27,28]. However, it is difficult to deploy those advanced LiDAR sensors (HDL-64 or VLS-128) for infrastructure-based LiDAR due to their expensive costs.

For better fitting of on-line point cloud filtering for low-channel infrastructure-based LiDAR, we proposed a novel slice-based projection filtering (SPF) algorithm. The aim of the method is to retain as much valuable information and remove as much worthless information as possible in a shorter processing time. The main work of this paper focuses on:(1)A 3-D point cloud is projected into 2-D polar coordinate points, and a slice-based projection filtering algorithm is proposed to reduce the point cloud data processing dimensions and computing time.(2)The point cloud is classified into four categories in a slice unit: Valuable object points (VOPs), worthless object points (WOPs), abnormal ground points (AGPs), and normal ground points (NGPs). Based on the point cloud classification results, the traffic objects and their surrounding information would be easily identified from an individual frame of point cloud.(3)A machine learning model based on a backpropagation artificial neural network (BP-ANN) is used to dynamically fit the road gradient or employing the inclination of the LiDAR sensor, as to improve the adaptability of the algorithm in a complex environment.

The rest of the paper is structured as follows: In Section 2, we introduce the LiDAR sensor and the type classification of data points. In Section 3, we give a detailed description of the assumptions and definitions proposed in this paper. In Section 4, we illustrate the detailed methods for background filtering. The experiments and validation analysis are presented in Section 5. In Section 6, we summarize the contributions of our work, as well as the perspectives on future work.

## 2. LiDAR Sensor and Points Classification

The working principle of LiDAR is to obtain the distance information of objects by firing and receiving laser beams. There are four important parameters: The number of laser channels, rotating frequency, vertical angle resolution, and horizontal angle resolution. These parameters determine the amount of points and the speed of the LiDAR sensor output point cloud. In this paper, we develop the point cloud filtering algorithm based on a 360° Velodyne VLP-16 LiDAR sensor that contains 16 laser channels. This sensor can detect ±15° in the vertical range, and the vertical angle resolution is 2°. We set the rotating frequency of LiDAR to 600 RPM and the corresponding horizontal angle resolution approximately 0.2°. Figure 1 demonstrates a selected frame of a raw 3-D LiDAR point cloud from the VeloView (Velodyne LiDAR, San Jose, CA, USA) software and field photo corresponding to LiDAR firing.

The LiDAR sensor outputs 3-D point cloud in spherical coordinates, which can be expressed as (d, ω, α), where d is the distance between a data point and the LiDAR, ω is the vertical angle fixed and given by the Laser ID (identity document number), and α is the horizontal rotation angle provided by the data packet of LiDAR. In this paper, we deal with the point cloud after projection in XYZ coordinates. The conversion relationship of the data point from spherical coordinates to XYZ coordinates is shown a s Figure 2 [29].

The spherical to XYZ conversion formulas are [30]:(1)$$\{\begin{array}{c} x=d\times \cos\left(\omega \right)\times \sin\left(\alpha \right)\\ y=d\times \cos\left(\omega \right)\times \cos\left(\alpha \right)\\ z=d\times \sin\left(\alpha \right) \end{array}.$$

A good filtering algorithm should retain as much valuable information as possible and remove as much worthless information as possible. That is, a good filtering algorithm should not simply filter all ground points. Instead, it should also retain all points with useful information for ground points, such as steps and curbs, which can also be used for road users’ path prediction. There are also worthless information points in object points, such as leaves, which have little impact on traffic. Therefore, we divided the points in a point cloud into four categories in this paper: Valuable object points (VOPs), worthless object points (WOPs), abnormal ground points (AGPs), and normal ground points (NGPs).

VOPs contain points that can affect traffic, such as tree trunks, road users, and lighting poles. As for WOPs, they mainly include points that cannot be used to judge objects and have no impact on traffic, such as noise and leaves. For NGPs, road surface points that can be normally used by vehicles are included. The algorithm proposed in this paper aims to remove WOPs and NGPs from the point cloud as much as possible. According to AGPs and VOPs, we can analyze pedestrians, vehicles, and their surrounding information (such as nearness to curbstones, the relative location of the road surface, adjacent position to a light pole), which are valuable inputs for microcosmic traffic behavior analysis, pedestrian zebra-crossing behavior analysis, driver or pedestrian intention prediction, trajectory tracking in traffic accident prevention, and other various intelligent transportation applications.

## 3. Definitions and Assumptions

### 3.1. Definition of Slice

Figure 3 shows the schematic diagram of LiDAR scanning objects in a data frame of the point cloud. In the point cloud, the point a and b are the points firing on the ground by LiDAR. The point c, d, and e are the points firing on the front object $O_{1}$. The point c, d, and e are within the same vertical block and form a slice within object $O_{1}$. Point f and g are the points firing on the rear object $O_{2}$. Point f and g are within the same vertical block and form a slice within object $O_{2}$. The point $c^{\prime }$, $e^{\prime }$, $f^{\prime }$, and $g^{\prime }$ are the points of the point c, e, f, and g mapping to the horizontal plane, respectively. As shown in Figure 3a,b, one object is composed of several slices in the same vertical plane. A vertical plane contains all the objects’ slices, which are in the same direction, as shown in Figure 3c. We define that a slice is a point set containing the points firing on the same object and within the same vertical block.

N is defined as the amount of points contained in a slice, and if N>1. If N=1, this indicates that it is a noise point or a ground point of LiDAR. The slice is an invalid slice that does not have enough information for further processing.

As shown in Figure 4a, the point α and β are the LiDAR data point in the same slice. The point $\alpha^{\prime }$ and $\beta^{\prime }$ are the points of the point α and β mapping to the XY horizontal plane, respectively. $\Delta D_{\alpha }^{\beta }$ is the spatial distance between point α and β. $\Delta R_{\alpha }^{\beta }$ is the distance between point $\alpha^{\prime }$ and $\beta^{\prime }$ in the XY horizontal plane. θ is the inclination angle of the object to the XY horizontal plane. $R^{\alpha }$ and $R^{\beta }$ are the distance of point $\alpha^{\prime }$ and $\beta^{\prime }$ to the ordinate origin, respectively. Their relationship can be formed as:(2)$$\Delta R_{\alpha }^{\beta }=\Delta D_{\alpha }^{\beta }\times \cos\left(\theta \right)=\left|R^{\alpha }-{\;R}^{\beta }\right|.$$

Generally speaking, objects, such as people, vehicles, and buildings, are vertically aligned with the ground. It means that the object’s inclination angle θ goes to 90°. Then:(3)$$\Delta R_{\alpha }^{\beta }=\Delta D_{\alpha }^{\beta }\times \cos\left(\theta \right)\approx 0.$$

As shown in Figure 4b, $\Delta D_{O1}^{O2}$ is the minimum distance between object $O_{1}$ and object $O_{2}$. There always have a social distance, safety distance, or viewing distance between pedestrians, vehicles, trees, and buildings. It can be deduced that:(4)$$\Delta R_{\alpha }^{\beta }<\Delta D_{O1}^{O2}.$$

Therefore, a threshold value $\Delta R_{th}$ can be found to distinguish whether the point α and β belong to the same slice:(5)$$\Delta R_{\alpha }^{\beta }<\Delta R_{th}<\Delta D_{O1}^{O2}.$$

Furthermore, the threshold value setting of $\Delta R_{th}$ will affect the accuracy of slice extraction within the object, particularly for the object, which is not perpendicular to the ground. For example, a too small $\Delta R_{th}$ may give rise to classification of the point α and β into different slices, when the point α and β actively belong to the same slice. On the contrary, a too large $\Delta R_{th}$ will result in the point α and β being classified into a slice, when the point α and β should be in different slices. From the experimental experiences, the threshold value of $\Delta R_{th}$ showed high accuracy and no obvious effect on the experiment results within the range of [0.1, 0.5]. In this paper, we set the threshold value as the median value of the range [0.1, 0.5], $\Delta R_{th}$ = 0.3 m.

### 3.2. Definition of Abnormal and Normal Ground Points

We assumed that there is a base ground plane (BGP), which is a reference plane and absolutely smooth. The curbstones and road obstacles can be regarded as the static objects on the BGP, and a pothole is also a type of object. As shown in Figure 5, if there is a curbstone on the BGP, the LiDAR sensor obtains the laser point as A; if there is no curbstone on the BGP, the LiDAR sensor obtains the laser point as B. According the classification of the point cloud in Section 2, point A belongs to AGPs and point B belongs to NGPs.

In Figure 5, h1 is the height of the curbstone, r is the horizontal distance between the curbstone and LiDAR sensor, and h2 is the installation height of the LiDAR. Δr is the horizontal distance between point A and B in the BGP. According to Equation (2), $\Delta r=\Delta R_{A}^{B}$ = $\Delta D_{A}^{B}\times \cos\left(\varphi \right)$.

### 3.3. Definition of Valuable and Worthless Object Points

In general, no matter the moving road users or the fixed tree trunks, they are all connected to the ground. We call them valuable objects. Road users are the main body of traffic, and other valuable objects will affect the trajectory of road users. Slices connected with or close to the ground are called valuable slices, and others are worthless slices. The corresponding points in slices are VOPs and WOPs. We define an area from the BGP to a certain height as a key region, as shown in Figure 6. The VOPs in the key region are called key valuable points.

In Figure 6a, the orange area is the key region, the red points are the key valuable points, and the blue points are other types of points. In Figure 6b, the red slices are the valuable slices and the gray slices are worthless slices. Most of the points in the key region are key valuable points, which are denser than other points in the point cloud within the key region. Based on this assumption, we can extract all the key valuable points. Then, all the object points can be classified into VOPs and WOPs.

## 4. Slice-Based Projection Filtering Algorithm

In this section, we propose the detailed steps of the slice-based projection filtering (SPF) algorithm, which makes up four major parts: Projection, extraction of valid slices, AGPs and NGPs separation, and VOPs and WOPs separation. Figure 7 shows the flowchart of the proposed method.

### 4.1. Projection

Compared with the 2-D plane, background filtering using 3-D space coordinates will take advantage of the rich characteristic information of the object. However, it will increase the calculation dimensions and computation complexity in dealing with laser points. For example, in 3-D-DSF background filtering, Wu [25] divided the 3-D space into continuous small cubes with the cube size at 0.1 m ×0.1 m ×0.1 m. For a 40 m ×40 m ×8 m scanning range, it needed to be divided into 400×400×80 cubes. There is only 400×400 cells to be analyzed if it taken into the 2-D plane. The loop count of computation with the 2-D plane is 1/80 of the 3-D space method. Therefore, we attempted to project the 3-D point cloud into the 2-D plane. The 2-D point cloud data can be represented by rectangular coordinates (x,y) and polar coordinates (r,α), as shown in Figure 2. These two coordinate systems take the LiDAR sensor as the origin of coordinates. The data point can be expressed as:(6)$$\{\begin{array}{c} x=r\times \sin\left(\alpha \right)=d\times \cos\left(\omega \right)\times \sin\left(\alpha \right)\\ y=r\times \cos\left(\alpha \right)=d\times \cos\left(\omega \right)\times \cos\left(\alpha \right)\\ z=d\times \sin\left(\omega \right) \end{array},$$
where x and y are the coordinate position within the *X*-axis and *Y*-axis, respectively, in rectangular coordinates; r is the distance between the point and LiDAR in the XY horizontal plane; and α is the horizontal rotation angle.

The process of projection in rectangular coordinates includes the multiplication of four trigonometric functions, while the projection in the polar coordinate system only requires the multiplication of one trigonometric function. Therefore, projection in polar coordinates can effectively reduce the calculation time and computation resources. Furthermore, the scanning trajectory will be a closed circle or ellipse after being projected when a laser beam in LiDAR is scanned on the ground plane.

There will be two y values on a certain x value when rectangular coordinates are used to represent this trajectory curve. If polar coordinates are used, r and α are a one-by-one correspondence. Hence, polar coordinates can be used to represent the trajectory curve with a simpler equation, which is more in line with the characteristics of the LiDAR data. In this paper, the 3-D point cloud is projected in a polar coordinate plane for analysis, as shown in Figure 8.

Figure 8 shows the comparison diagram of the point cloud before (Figure 8a) and after the projection (Figure 8b). We find that the height information of the point cloud is hidden after the projection. However, this will not affect the processing results, as the purpose of the projection in this paper is to extract the valid slices in the point cloud.

### 4.2. Valid Slices Extraction

The purpose of the valid slice extraction operation is to find the object points in the point cloud based on the properties of the slice. We used the matrix $C_{0}=\left[D,W,A\right]$ to represent the raw data of the LiDAR rotation output, where D, W, and A represent the column vectors that contain d, ω, and α, respectively, and each row corresponds to a point collected by the LiDAR sensor. We used matrix $C_{1}=\left[R,W,A\right]$ to represent the matrix after projecting matrix $C_{0}$. We used $C_{1}\left[flag\right]$ to obtain all row vectors in the $C_{1}$ satisfying the condition of the flag.

Suppose $P_{x}=\left(r_{x},\omega_{x},\alpha_{x}\right)$ is a certain row of $C_{1}$ and we need to find other points in the same slice with $P_{x}$, it is the simplest way to traverse all points in $C_{1}\left[\alpha =\alpha_{x}\right]$ and calculate the distance from $P_{x}$. However, there are many points in the point cloud, and it will take a long time to traverse all points, so the implementation efficiency of this method is very low. For this reason, this paper proposed a solution based on matrix operations to extract effective slices efficiently, as shown in Algorithm 1.
**Algorithm 1** Extracting Valid Slices1:**function** LABLEPOINTSINSLICES ([R,W,A])2:Sort A column and W column in ascending order for [R,W,A], obtain $\left[R^{\prime },W^{\prime },A^{\prime }\right]$3:Get the difference value between two elements in $R^{\prime }$ sequences, obtain delta4:FOR i=0 TO Count(delta)−1 DO5:IF $\mathrm{Abs}\left(delta\left[i\right]\right)<\Delta R_{th}$ THEN F1[i+1]=TRUE ELSE F1[i+1]=FALSE6:END7:F1[0]=FALSE8:Locate the min(ω) in $W^{\prime }$ and obtain its sequence number ID, set F1[ID]=FALSE9:TempVector=F110:FOR i=0 TO Count(TempVector)−2 DO11:F1[i]=(TempVector[i] ∥ TempVector[i+1])12:END13:F1[Count(TempVector)−1]=TempVector[Count(TempVector)−1]14:return F1 and TempVector15:**function** LABLESLICES(TempVector)16:Set the index of current labeled slice: currentID=017:temp= the number of points in current slice18:FOR i=0 TO Count(TempVector)−1 DO19:**IF**TempVector[i] == False **THEN**20:**IF** temp! = 0 **THEN**21:currentID=currentID+122:F2[(i−1−temp) TO (i−1)]=currentID23:temp=024:**ELSE**25:temp=temp+126:**return**F2

According to the definition and attributes of valid slice, the valid slices can be extracted based on two conditions:

Condition 1: If points belong to the same slice, these points have the same horizontal rotation angle.

Condition 2: The horizontal distance of two points $\Delta R_{\alpha }^{\beta }$ in the same slice is less than the threshold value of $\Delta R_{th}$.

Therefore, we suppose P1, P2, and P3 are three points that are all contained in $C_{1}$. If P1 and P3 belong to the same slice, P1 and P2 also belong to the same slice, then it is clear that P2 and P3 also belong to the same slice. Thus, we only need to determine the distance between two adjacent laser points within the setting horizontal rotation angle.

We can obtain the matrix $C_{1}^{\prime }=\left[R^{\prime },W^{\prime },A^{\prime }\right]$, where two adjacent rows are also adjacent in real space after sorting the rows of matrix $C_{1}$ by A and then by W. Then, it is easy to get the distance difference of all adjacent points to LiDAR. Finally, we can reasonably set the threshold value of $\Delta R_{th}$ to extract all points in the point cloud that belong to the valid slices. In algorithm 1, matrix F1 is used to mark all points, where true indicates that the point belongs to a certain slice and false indicates that the point is a single point.

There are one or more single points between the different slices. Based on the differences, the slice can be differentiated from and labeled. In the proposed algorithm, we used the matrix F2 to mark different slices. Within the matrix F2, 0 indicates that the point is a single point; if the number is greater than 0, it indicates the index of the slice that the point belongs to.

By the experiments, the runtime of the proposed algorithm is about 18 ms compared to the traversing method, which needed 1.5 s to extract the valid slice. Therefore, the proposed algorithm of this paper is more efficient.

### 4.3. Abnormal and Normal Ground Points Separation

The surface, slope, curbstones, road obstacles, and other information of the ground are also very useful for trajectory predictions of road users with the consideration of the edge of different function areas of the road. The purpose of the AGPs and NGPs separation operation is to abstract all the ground points that could impact the road users.

#### 4.3.1. Ground Point Extraction

After extracting the valid slices from the point cloud, the remaining points are a single point set. The single point set is composed of the ground points, noise points, and edge points of objects. In the LiDAR scanning range, the ground points are closer to the BGP. Therefore, we can obtain the ground points by setting the condition as follows:(7)$$IF\;h_{X}<Z_{X}+H_{gp}\;THEN\;X\;\mathrm{is}\;\mathrm{the}\;\mathrm{ground}\;\mathrm{point},$$
where $h_{X}$ is the height of point X; $Z_{X}$ is the coordinates of the point X on the Z-axis, where the horizontal plane is the BGP; and $H_{gp}$ is the threshold of ground points.

In this paper, the threshold value $H_{gp}$ was based on the height between the point cloud and BGP. We divided 0–1 m into 100 parts on average and counted the amount of points in each part. The height distribution of the point cloud is shown in Figure 9.

As shown in Figure 9, there is a critical point (0.26, 40) in Figure 9. The point cloud with 0–1 m is divided into two parts at the critical point (0.26, 40). The height of most ground points is less than 0.26 m. The height of most objects’ points is higher than 0.26 m. Thus, we set $H_{gp}=0.26$ in this paper.

#### 4.3.2. Separation Method

In the absence of AGPs, the projected NGPs should be a closed curve track, noted as L1. When the projection plane of the LiDAR point cloud is parallel to the ground, the trajectory is a circle. The trajectory is a curve similar to an ellipse when LiDAR has an angle from the ground. When there is an AGP at a horizontal angle α, the point should deviate from the curve L1. Therefore, we can fit the trajectory curve L1, and use it to decide whether a point P in the ground points belongs to the NGPs or AGPs according to the distance from the curve L1. The judgement condition is:(8)$$IF\;Abs\left(R_{P}-R_{Predict}|\alpha =\alpha_{P}\right)\leq P_{th}\;THEN\;P\in NGPs\;ELSE\;P\in AGPs,$$
where $R_{P}$ is the actual distance between the projected point P and LiDAR; $R_{Predict}|\alpha =\alpha_{P}$ is the distance predicted by the curve L1 at the horizontal roatation angle $\alpha_{P}$; Abs(∗) is the absolute function; and $P_{th}$ is the threshold value to differentiate AGPs and NGPs; the range of $P_{th}$ is:(9)$$0<{\;P}_{th}<Max\left(P_{th}\;\right)=\frac{R_{P}*h1}{h2-h1}=R_{P}\times K,$$
where K is a coefficient, and K=h1/(h2−h1); h1 is the height of the point; and h2 is the installation height of the LiDAR.

We selected the data of laser channel 0 with the smallest vertical angle to train the trajectory curve L1, as the points where the laser of this channel fell on BGP were the closest to the LiDAR, and contained the most ground information. In addition, we selected a frame of the point cloud including more ground information, so as to fit the curve more accurately. After fitting the curve L1, the curves can be applied to the point cloud data of all other frames.

The α and r of all points obtained by a single channel laser beam in a scanning period are one-by-one corresponding. As shown in Figure 10, with α taken as the horizontal axis and r as the vertical axis, the processes of the fitting curve can be described as the following steps:

**Step 1**: We used a quadratic equation to fit the ground points detected by the LiDAR laser channel 0 by least squares, firstly:(10)$$fit\;function1=\lambda \;\cdot r^{2}+b\cdot \;r+c,$$
where r is the distance between the point and LiDAR in the XY horizontal plane as mentioned above; and λ, b, and c are regression coefficients, which can be obtained by least squares regression.

As shown in Figure 10a, the green scatter is the ground point detected by the laser channel 0, noted as GP0. The orange curve is the fitting curve of the ground points, noted as L0. It can be seen that there are some points far below the fitting curve L0. These are noise points or AGPs in GP0.

**Step 2**: Base on the fitting curve L0 fitted by the laser channel 0, we used L0 to filter the points detected by LiDAR laser channel 1 to channel 15. With the same horizontal angle α, if the points detected by the LiDAR laser channel 1 to channel 15 are far below the corresponding point in L0, these are removed before stepping into the next step.

**Step 3**: After filtering out all the noise points or AGPs in step 1 and step 2, the remaining points are the NGPs. We used the remaining points to fit the final curve L1 by the Sinusoid function as:(11)fit function2=A·sin(δ·α+ϑ)+γ,
where α is the horizontal rotation angle of the point as mentioned above; and A, δ, ϑ, and γ are the regression coefficients, which can be obtained by least squares regression.

As shown in Figure 10b, the green scatters are the NGPs. The orange curve is the final fitting curve L1. It can be seen that some blue scatters are far below the fitting curve L1. These are noise points or AGPs within the raw data. The remaining points nearly around the fitting curve L1 map well with the NGPs. Most of the blue points nearly around the fitting curve L1 overlap with the corresponding NGPs. It means that the fitting curve L1 can effectively distinguish the AGPs and NGPs.

#### 4.3.3. Separation Results

After obtaining the trajectory equations of all laser channels, we can distinguish AGPs and NGPs in all ground points by setting the threshold value $P_{th}$ reasonably. In practical applications, the values of h1 and h2 are fixed. According Equation (9), the max($P_{th}$) will change as the actual distance $R_{P}$ is changing. As to distinguish the AGPs and NGPs accurately, the threshold value $P_{th}$ should change as the as the actual distance $R_{P}$ changes. In the experiment, we set h1 equal to 0.26 m based on the height of the curbstones and h2 equal to 1.8 m based on the installation height of the LiDAR. According K=h1/(h2−h1) in Equation (9), K≈0.1688. Thus, $0<{\;P}_{th}<R_{P}\times 0.1688$. The experiments results showed that a too small threshold value $P_{th}$ will give rise to recognition of the NGPs as AGPs when the points are near the LiDAR sensor. On the contrary, a too large threshold value $P_{th}$ will result in the AGPs being identified as NGPs when the points are far away from the LiDAR sensor. In this paper, we used the median value of the range $\left(0,\;R_{P}\times 0.1688\right)$, setting $P_{th}=R_{P}\times 0.0844$. The experiment result is shown in Figure 11.

As shown in Figure 11a, the blue scatters are NGPs, and the orange scatters are AGPs. It can be effectively distinguished by the proposed method. If there are a large number of AGPs and a small number of NGPs in a certain area, it can be deduced that this area is a non-driving area due to too many obstacles within this area. The gray areas are non-driving areas as shown in Figure 11a. Figure 11b shows the experimental location in the Baidu Map and the field scene scanned by the LiDAR sensor in the experiment.

### 4.4. Valuable and Worthless Object Points Separation

We can obtain the object points in the point cloud after extracting the valid slice operation. However, not all object points are useful for traffic behavior analysis. Therefore, the separating VOPs and WOPs operation is required to find the points from all object points, which are useful for traffic behavior analysis.

#### 4.4.1. Key Region Optimization

If the projection plane of the LiDAR point cloud is parallel to the BGP, the Z information of the point cloud can be directly used to obtain the key region. In fact, there may be a slope in the BGP. In this situation, it is hard to ensure the projection plane remains parallel to the BGP. Therefore, if there is an inclination angle between the BGP and the projection plane, it will be difficult to obtain the accurate key region by using only the Z information of the point cloud. The error increases with the distance from the LiDAR sensor, as shown in Figure 12.

As shown in Figure 12a, point A and point B are formed by LiDAR laser beams falling on the BGP. The projected points are $A^{\prime }$ and $B^{\prime }$, respectively. ∂ is the inclination angle between BGP and the projection plane. When another laser beam falls on point X on the BGP, the Z-coordinate of point X should be:(12)$$Z_{X}=\tan\left(\partial \right)\times R_{X}=\frac{Z_{B}-Z_{A}}{R_{B}-R_{A}}\times R_{X},$$
where $R_{A}$, $R_{B}$, and $R_{X}$ are the horizontal distance from the point A, B, and X to the LiDAR sensor, respectively; and $Z_{A}$, $Z_{B}$, and $Z_{X}$ are the distance deviation in calculation the Z-coordinate of point A, B, and X, respectively.

Clearly, if there is an inclination angle between the BGP and the projection plane, the ∂ is changing for different points within the point cloud, particularly for the object points. It requires at least two channel laser points to calculate the inclination angle between the BGP and the projection plane.

According to the LiDAR user manual and programming guide [29], the laser ID 0 and 2 are the two laser beam closer to the ground. In the experiment, these two laser beams generate more ground points than the other laser beam in LiDAR. In this paper, the NGPs generated by laser ID 0 and 2 are used to calculate the inclination angle ∂ between BGP and the projection plane. The NGPs are obtained based on the process results of Section 4.3, as shown in Figure 12b.

In the LiDAR data packet, we can obtain the XYZ coordinates of the point and its horizontal rotation angle. Here, $A\left(x_{A},y_{A},z_{A}\right)$ is the coordinate information of point A obtained by the laser ID 0, $B\left(x_{B},y_{B},z_{B}\right)$ is the coordinate information of point B obtained by the laser ID 2, and point A and point B have the same horizontal rotation angle. According to Equation (12) and Figure 12a, the inclination angle ∂ between BGP and the projection plane can be calculated as:(13)$$\tan\left(\partial \right)=\frac{z_{B}-z_{A}}{\sqrt{{\left(x_{B}-x_{A}\right)}^{2}+{\left(y_{B}-y_{A}\right)}^{2}}}.$$

All the point pairs in Figure 12a were used to calculate tan(∂), and the calculation results are shown in Figure 13a, where the green scatters is the value of tan(∂) at different horizontal rotation angles (0–2π).

As shown in Figure 13a, there is noise in the calculation results, which is caused by the errors of the sensor itself and the uneven ground. The continuous curve of the tan(∂) value cannot be obtained directly as the points used to calculate it are discrete, or there is a shied on the road.

To improve the flexibility of the algorithm, it is necessary to fully consider the inclination angle information between the BGP and the projection plane. In addition, the road surface condition also should be considered. To adapt to different road surface conditions and employ inclination, we attempted to use a machine learning algorithm to obtain accurately the inclination angle between the BGP and the projection plane and identify the key region. A machine learning model based on the backpropagation artificial neuron network (BP-ANN) was developed to fit the scatter of thte calculated tan(∂) to obtain an accurate and continuous actual tan(∂).

The BP-ANN is a multilayer feedforward neural network composed of an input layer, a hidden layer, and an output layer, as shown in Figure 13b. In practical applications, the road pavement conditions within the scanning range of LiDAR are different. Thus, in the different LiDAR scanning horizontal rotational angles α, the inclination angle ∂ may be different. In the BP-ANN, we input the horizontal rotational angle α to get the corresponding road pavement inclination ∂ with the output of tan(∂) by the BP-ANN-based machine learning model. It means that tan(∂)=Func(α) and Func(∗) is obtained by the BP-ANN model. The training data set of α and tan(∂) were the points on the BGP with the same horizontal rotational angle α, using Equation (12) or Equation (13) to calculate the corresponding tan(∂). Therefore, the input layer and output layer contained only one neuron each as only one variable was needed for each layer. The variables α and tan(∂) were used for the input layer and the output layer, respectively. The number of neurons of the hidden layer was set as 32. We used the activation function for neurons in the hidden layer for the function that could converge faster than the sigmoid function, while the neuron in the output layer did not use any activation function to avoid gradient explosion. During the model training period, the mean square error function was used for the loss function, the optimizer was Adam, and the batch size was set as 32. The fitting curve of tan(∂) is shown as the red line in Figure 13a.

In addition, we also used three other algorithms to fit the tan(∂) curve. A machine learning model based on long short-term memory (LSTM) was used to fit the tan(∂) curve [31]. The complete network model is shown in Figure 13c, in which each LSTM cell contains 15 neurons. The time-step of the model was set to 2, due to the tan(∂) being predicted by keeping the memory of several previous position points when both the LiDAR and the BGP are fixed. Then, the memory of the changing tan(∂) was transferred between two adjacent batches until the internal state of the model was reset. During the model training period, the mean square error function was used for the loss function, the optimizer was Adam, and the batch size was set as 8. After the completion of the model, we set the time-step to 1 for the prediction, and the obtained continuous tan(∂) curve is shown as the blue line in Figure 13a.

The second method was the Ridge regression [32], where the coefficient of the penalty term was set as 0.5. The fitting curve of tan(∂) is shown as the yellow line in Figure 13a. The last method was support vector regression (SVR) [33], where the radial basis was used for the kernel function. The fitting result of the tan(∂) curve is shown as the orange line in Figure 13a.

From the fitting results, we can see that the fitting results based on LSTM are the best in the range of [1.5π,2π]. The fitting curve of tan(∂) based on BP-ANN is similar to LSTM. The loss curve generated by the mean square error function with the LSTM model dropped faster than the BP-ANN model. However, the loss values were almost the same after 100 epochs, as shown in Figure 13d. In practical applications, the LSTM-based machine learning model is much more complex than the BP-ANN machine learning model. Therefore, we chose the simpler BP-ANN machine learning model to obtain the accurate inclination angle between the BGP and the projection plane.

After identification, there is an inclination angle between the BGP and the projection plane, and after obtaining the accurate tan(∂) in the corresponding horizontal rotation angle, it is easy to identify a point within the point cloud that belongs to the key region by:(14)$$z<Z_{X}-H_{LiDAR}+H_{KR},$$
where z is the Z-coordinate output by the LiDAR sensor; $H_{LiDAR}$ is the installation height of the LiDAR sensor; $H_{KR}$ is the height of the key region; and $Z_{X}$ is the distance deviation, which can be calculated according to Equation (12).

In China, the average height of male adults is about 1.71 m, and it is about 1.6 m for female adults [34]. The height of small cars is generally within 1.6 m and 1.8 m. Road users are the main research object in the field of transportation. Therefore, in the experiments, we set the value of $H_{KR}$ as 1.5 m, slightly lower than the general height of road users. Within this height range, most road users can be included, and most roadside leaves can be excluded.

#### 4.4.2. Valuable Object Points Extraction

The key valuable points are more point intensive than other types of points after projecting all points in the key region. Therefore, the DBSCAN (density-based spatial clustering of applications with noise) [35] algorithm can be used to extract the key value points from the key region. With DBSCAN, one object can be joined in the clustering area if the point density is higher than or equal to a pre-defined minimum number of points (MinPts), and the distance to the center is less than a pre-defined epsilon (ε). In the experiments, we set Minpts=3 and ε=0.008 to cluster the projected points in the key region. The key value point cluster results are shown as Figure 14a.

As shown in Figure 14a, the green scatters are the key valuable points obtained from the point cloud in Figure 8. The red scatters are sparse points that belong to WOPs. According to Algorithm 1, we can obtain the matrix F2, which contains the index of all valid slices. Therefore, we can find all the index numbers contained in the key valuable points and then find all the VOPs by combining with the matrix F2.

Figure 14b is generated by the NGPs that were obtained based on the process methods of Section 4.3, and has the same raw point cloud of Figure 14a. According the fitting curve of tan(∂), we can infer that the height of the left side is higher than the right side in Figure 14. If only the Z coordinate is used for the identification of the key region, we will obtain less area for the left side and more in the right side.

Table 1 shows the numbers of VOPs obtained in the left and right sides of Figure 14, with or without the fitting of the inclination angle between the BGP and the projection plane.

As shown in Table 1, on the left side of the key region, the number of VOPs obtained after fitting the inclination angle was more than that without fitting, and it was an adverse result on the right side. This phenomenon demonstrates that greater accuracy in the key region can be found after fitting the angle. In addition, the number of VOPs with fitting was more than that without fitting, which shows that the greater the accuracy in the key region, the more VOPs that will be obtained.

Figure 15 shows the AGPs and VOPs extraction results with or without the fitting of the inclination angle between the BGP and the projection plane.

As shown in Figure 15a, the VOPs within the green ellipse region are missing in Figure 15b. Furthermore, the inclination angle between the BGP and the projection plane not only affects the extraction result of VOPs but also affects the extraction result of AGPs. As shown in Figure 15, the AGPs within the orange ellipse regions in Figure 15a are missing in Figure 15b, which does not have fitting of the inclination angle. This is due to the left side being higher than the right side as discussed above in Figure 14, and a large number of ground points are far from LiDAR on the left side being identified as individual single points, which belong to the objects in Figure 15b. In addition, as shown in Figure 15b, some single points within the red ellipse region, which belong to the objects on the right side, are identified as the AGPs. Therefore, it is necessary to fit the inclination angle to improve the accuracy in the identification of the category of points.

## 5. Experiments and Validation Analysis

### 5.1. Filtering Results

The first experiment was designed to illustrate the obtained filtering results. We evaluated the performance of the proposed method of this paper and compared it to a popular RANSAC algorithm by Behley [23] and the 3-D-DSF algorithm provided by Wu [25]. For the 3-D-DSF approach, the side length of the cube was 0.1 m.

The point cloud data were collected by the Velodyne VLP-16 LiDAR sensor at Central Avenue, Nanling campus, Jilin University. The location of the LiDAR sensor is latitude 43.85 n, longitude 125.33 w. Approximately 1200 frames of point cloud were collected and used as input data in the experiments. The background point filtering results of the three algorithms are shown in Figure 16.

Figure 16a shows the raw point cloud before processing. Figure 16b shows that the RANSAC algorithm can only filter the ground points from the raw point cloud. More processing stages are required to remove the worthless points from the point cloud by other methods. As shown in Figure 16c, there are still some noise points left after the background point filtering. We needed to remove noise points using clustering methods, such as k-means and DBSCAN, which increases the execution time of the algorithm [36]. Furthermore, the 3-D-DSF algorithm ignores certain useful information generated by fixed objects, such as curbstones and tree trunks, which can affect the trajectory of a road user in summer in tropical areas.

Figure 16d shows the filtering result by the proposed algorithm SPF. Different colors in the figure present different objects clustered by DBSCAN. From the perspective of filtering quality, the proposed algorithm was able to effectively extract useful points in the point cloud, including the AGPs that belong to the ground and the VOPs that belong to the objects. Through AGPs and VOPs, we can not only obtain the information of all road users in the point cloud but also the environment information that can affect the trajectory of the road users. Therefore, analyzing the data after filtering by the proposed algorithm can help to perceive the traffic information more comprehensively, accurately, and quickly.

To compare the filtering results in more detail, we used the amount of the point cloud before and after filtering to evaluate the algorithms. The experimental results are compared in Table 2, and Figure 17 and Figure 18. The raw point cloud, raw VOPs, and raw AGPs in the experiments are shown in Figure 16a, Figure 17a, and Figure 18a, respectively.

As shown in Table 2, the 3-D-DSF algorithm can filter out 96.7% of the points from the point cloud. The VOPs and AGPs excluded percentages were 84.5% and 96.8%, respectively, as shown in Figure 17c and Figure 18c, respectively. However, the 3-D-DSF algorithm filtered out too much valuable information. Thus, the amount of valuable points (VOPs and AGPs) after background point filtering was only 956, which was the minimum. However, it still contained some noise after the background point filtering by 3-D-DSF. The percentage of noise point in the valuable points was about 4.29%.

The background point filtering percentage of RANSAC was only 15.9% in the raw point cloud filtering. The VOPs and AGPs excluded percentages were 11.7% and 7.0%, respectively, as shown in Figure 17b and Figure 18b, respectively. This showed that the RANSAC algorithm retained too much worthless information. This is why the amount of valuable points after background point filtering was only 6321, which was the maximum. However, it needed to further exclude about 74% of the worthless points after the raw point cloud filtering. The percentage of valuable point was only 26% after background point filtering by RANSAC, which was the minimum.

The SPF algorithm, the proposed method in this paper, filtered out 80.1% of the points from the raw point cloud. The VOPs and AGPs excluded percentages were 20.0% and 14.8%, respectively, as shown in Figure 17d and Figure 18d, respectively. It could exclude all of the worthless points in the raw point cloud background filtering. The percentage of valuable point was almost 100% after background point filtering by SPF in the raw point cloud filtering, which was the maximum.

In terms of the filtering quality and accuracy for further processing in identifying the traffic object, the SPF algorithm was able to retain more valuable information and remove more worthless information for further processing in the stage of traffic object identification and trajectory tracking. In this respect, we concluded that the SPF algorithm performed better than RANSAC and 3-D-DSF.

### 5.2. Runtime

The execution time of the algorithm plays an important role in practical applications, especially in area traffic behavior and safety analysis. Therefore, the experiment was designed to evaluate the running performance of the three algorithms. The three algorithms were implemented in Python3.5 and were deployed on a desktop computer equipped with an Intel Core i7-7500U CPU (2.70 GHz), with 8 GB of RAM, and with a Windows 10 operating system. The raw point cloud was collected by the Velodyne VLP-16 LiDAR sensor located at Central Avenue, Nanling campus, Jilin University. The raw point cloud including about 1200 data frames was used as the data input of the three algorithms. The performance results are shown in Table 3.

As shown in Table 3, the average background point filtering time was about 51 ms per frame by the SPF algorithm. The running times of RANSAC and 3-D-DSF were 198 and 113 ms per frame, respectively. In addition, the 3-D-DSF algorithm needed multi-frames to perform the algorithm. It would not be suitable for on-line intelligent transportation applications.

## 6. Conclusions and Discussion

A LiDAR sensor has advantages in obtaining the high-resolution fine-grained position, direction, speed, and trajectory data of vehicles and pedestrians. However, the point cloud generated by a LiDAR scan contains not only road user points but also buildings, trees, and ground points. To speed up the micro-level traffic data extraction from the LiDAR sensor, removing the background points using a suitable background filtering algorithm is necessary. In this paper, we developed a novel method of background point filtering for low-channel infrastructure-based LiDAR.

First, we projected a 3-D point cloud into the 2-D polar coordinate points for filtering to reduce the data processing dimensions and computing time. Secondly, we divided the points in the point cloud into four categories: AGPs, NGPs, VOPs, and WOPs. Then, we used the slice as a unit to analyze the point attributes, and to separate and extract the VOPs and AGPs from a single frame of the point cloud. To dynamically fit different road conditions and situations, such as different road gradients and different LiDAR scanning angles, a machine learning method based on the artificial neural network was used to improve the adaptability of the algorithm in complex environments.

Finally, compassion was done to verify the efficiency and accuracy of the method in this paper. Compared to the methods of RANSAC and 3-D-DSF, the proposed algorithm of this paper retained more valuable points, removed more worthless points, and required fewer computational sources. In addition, there were less parameters required for calibration in the proposed algorithm. We have the following major findings:(1)Projecting 3 = D point cloud into 2-D polar coordinates can reduce the computation complexity and memory resource in the point cloud loop processing during the background point filtering. The processing time using 2-D polar coordinates is about one quarter of the processing time using the 3-D point cloud directly.(2)Using the vertical slice unit, which is different from the horizontal slice unit mentioned in the the state-of-the-art literature, can effectively classify the point cloud into VOPs, WOPs, AGPs, and NGPs. It can further remove the worthless points in the background point filtering. The accurate and excluding efficiency is about 5 times that is not based on the vertical slice unit.(3)In practical applications, road pavement conditions and LiDAR installation statuses need to be considered, such as the road gradient and LiDAR sensor inclination. Therefore, a BP-ANN machine learning model was proposed to fit the inclination angle between the BGP and the projection plane to improve the flexibility of the background point algorithm in a complex environment.

However, the present study and its methodological approach are not free from limitations. The first limitation is that the slice-based background point filtering algorithm did not consider the object far away from the LiDAR sensor, which can be obtained from only one single point. Based on the vertical slice unit rule, these points presenting the objects are considered as noise in the algorithm and filtered out. Therefore, it needs to identify the effective filtering range of the proposed algorithm. A second limitation is that the non-driving area far away from the LiDAR sensor cannot be detected accurately, due to a lack of reference points, and the threshold used to differentiate point types increases as the distance increases in the present study. However, the non-driving area is important for pedestrian trajectory tracking and intention prediction in intelligent transportation applications. The third limitation is that the present study did not consider the weather conditions. The proposed algorithm only evaluated during the daytime on a sunny day. However, the bright day and the black night have a different impact on the laser refection intensity of the LiDAR sensor. Snow and rain falling on the ground will influence the accuracy of ground point identification. A fourth limitation is the efficiency and accuracy of traffic object identification and traffic object trajectory tracking from the VOPs and AGPs. The effectiveness and necessity to retain AGPs for assistant identification of the traffic object and trajectory should be further evaluated. The final limitation is that the present study used low-density point cloud from the Velodyne VLP-16 low-channel LiDAR sensor to evaluate the efficiency and accuracy of the proposed algorithm. The high-density point cloud data from a higher channel LiDAR sensor, such as Velodyne VLP-32C and HDL-64, or other manufacturers’ LiDAR product models with the same or higher channel, should be used to evaluate the proposed algorithm to test and validate the efficiency, accuracy, and portability of the proposed algorithm in different LiDAR products. For further studies, such problems and factors should be considered and added in the modeling, testing, and validation of the model.

## Figures and Tables

**Figure 1 sensors-20-03054-f001:**
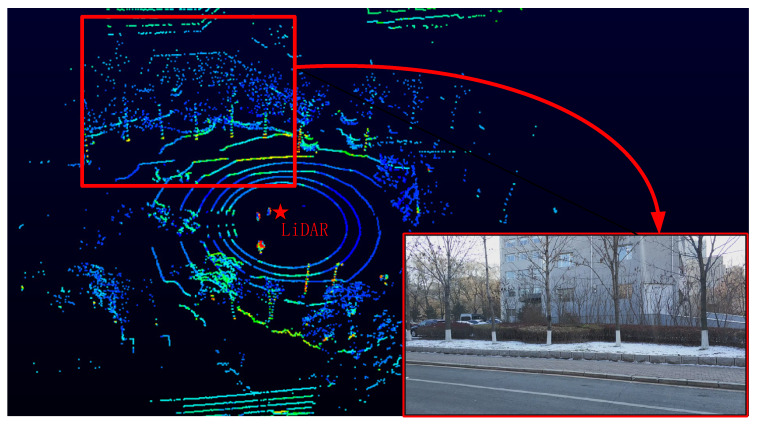
A frame of a light detection and range (LiDAR) point cloud displayed by the VeloView and the corresponding field photo.

**Figure 2 sensors-20-03054-f002:**
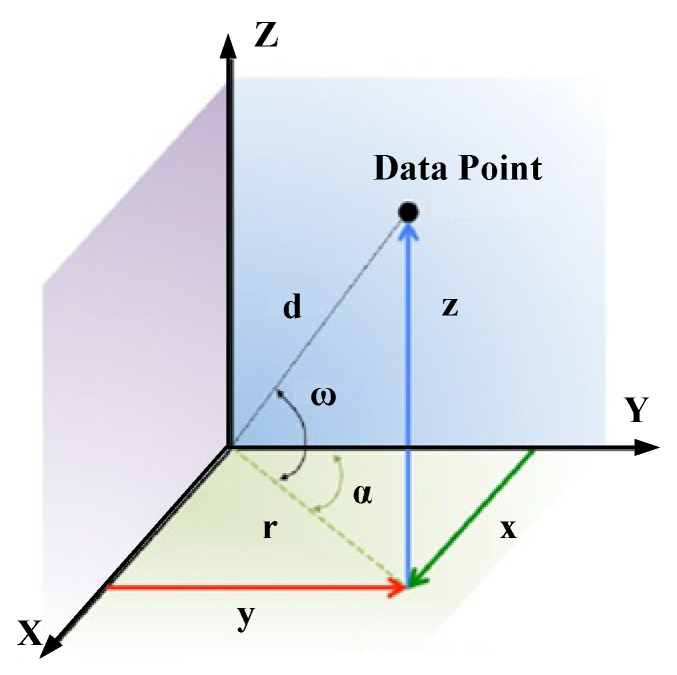
The conversion relationship of the data point from spherical coordinates to XYZ coordinates.

**Figure 3 sensors-20-03054-f003:**
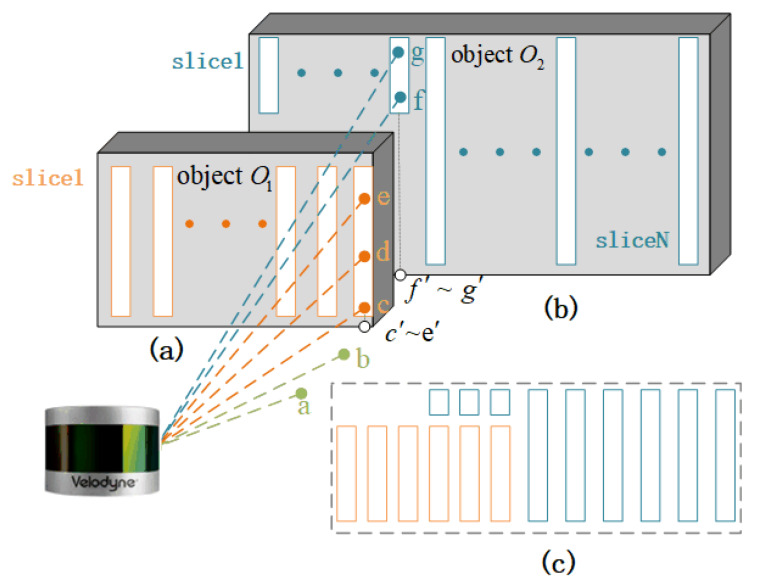
The schematic diagram of a LiDAR slice. (**a**) Points and slices within object $O_{1}$. (**b**) Points and slices within object $O_{2}$. (**c**) Slices and objects within a vertical plane.

**Figure 4 sensors-20-03054-f004:**
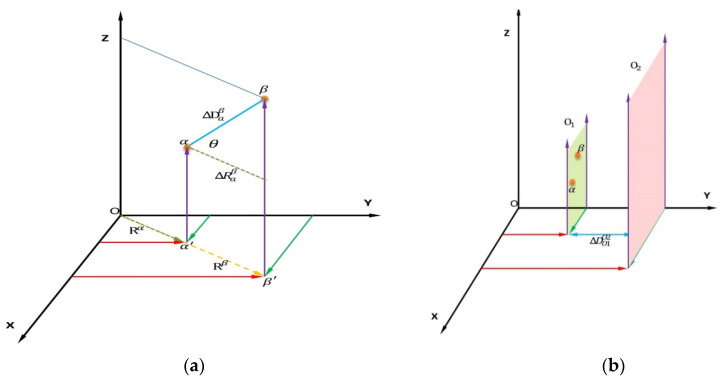
The points’ relationship in a slice and between objects. (**a**) The points’ relationship in a slice. (**b**) The points’ relationship between objects.

**Figure 5 sensors-20-03054-f005:**
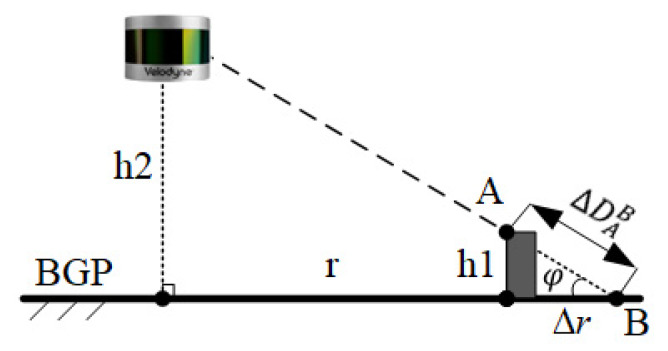
Schematic diagram of distinguishing abnormal ground points (AGPs) and normal ground points (NGPs).

**Figure 6 sensors-20-03054-f006:**
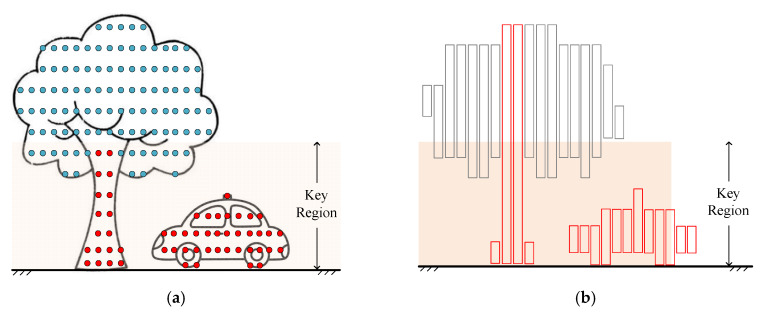
Schematic diagram of separating valuable object points (VOPs) and worthless object points (WOPs). (**a**) The schematic diagram showing the key region and key valuable points in the key region; (**b**) The schematic diagram of obtaining all valuable slices in the point cloud based on the key valuable points in the key region.

**Figure 7 sensors-20-03054-f007:**
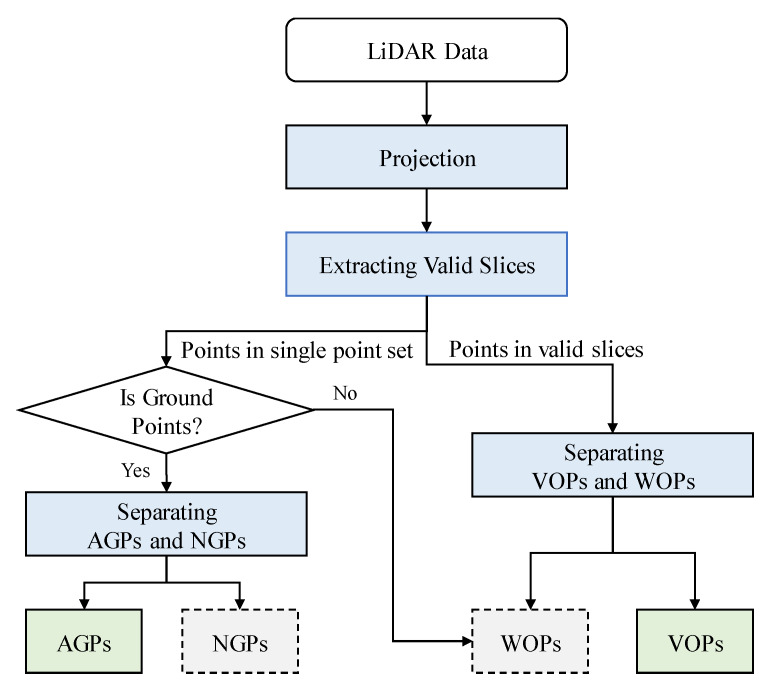
Flowchart of the slice-based projection filtering algorithm.

**Figure 8 sensors-20-03054-f008:**
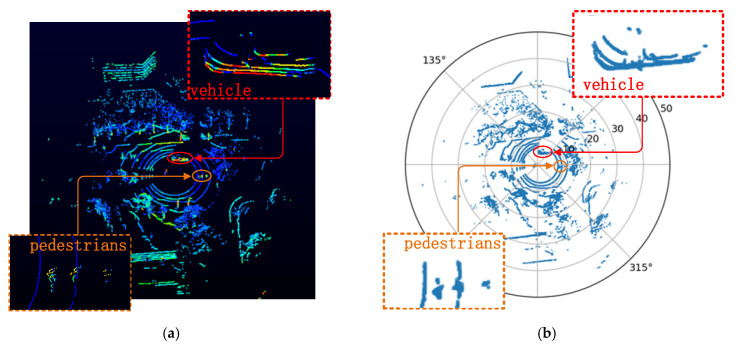
The point cloud display in different coordinates. (**a**) The original point cloud displayed in the VeloView software. (**b**) The result of the projecting point cloud.

**Figure 9 sensors-20-03054-f009:**
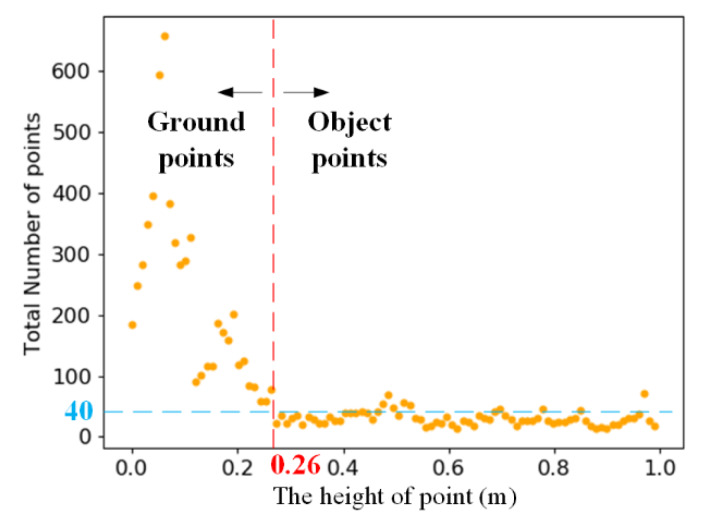
The height distribution of the point cloud.

**Figure 10 sensors-20-03054-f010:**
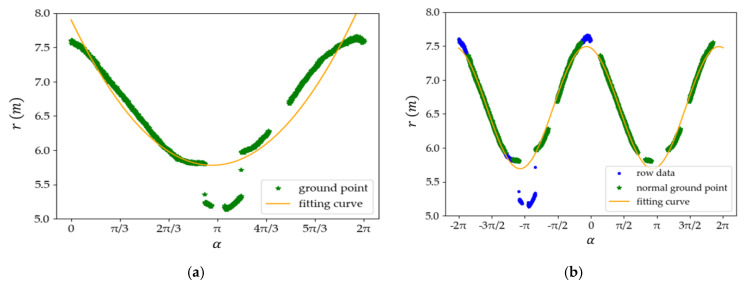
The processes of the fitting curve L1. (**a**) Fitting the L0 curve in step 1; (**b**) Fitting the final L1 curve in step 3.

**Figure 11 sensors-20-03054-f011:**
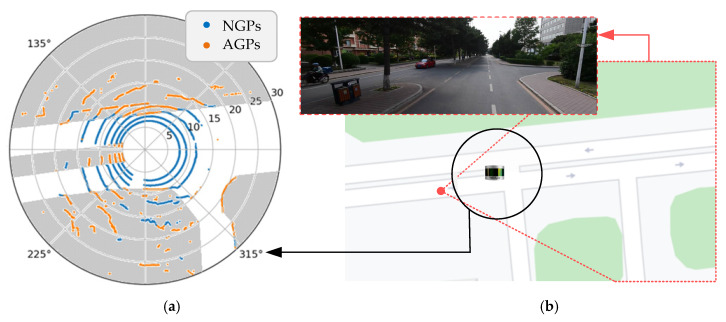
Abnormal ground points (AGPs) and normal ground points (NGPs) separation result. (**a**) The distribution of AGPs and NGPs in the polar coordinate. (**b**) The field scene of the LiDAR sensor scanning.

**Figure 12 sensors-20-03054-f012:**
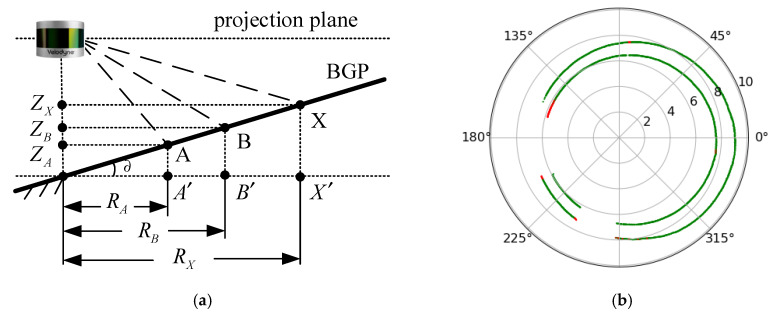
Calculating the slope between the base ground plane (BGP) and the projection plane. (**a**) A mathematical model using LiDAR to calculate the slope; (**b**) The points used to calculate the slope.

**Figure 13 sensors-20-03054-f013:**
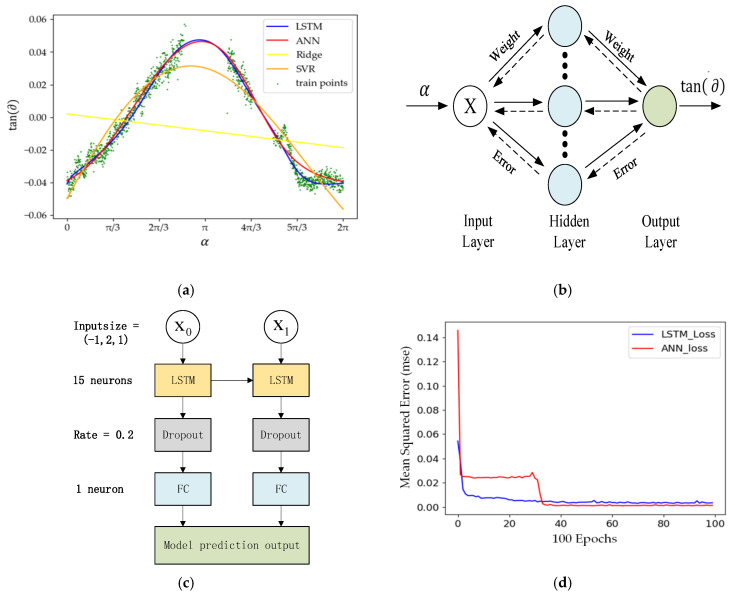
The inclination angle between the BGP and the projection plane obtained by the machine learning model. (**a**) The fitting results of the tan(∂) curve by different machine learning models. (**b**) The schematic diagram of a backpropagation artificial neuron network (BP-ANN) machine learning model. (**c**) The schematic diagram of a long short-term memory (LSTM) machine learning model; (**d**) The loss curves generated by the mean square error function with the LSTM model and BP-ANN mode.

**Figure 14 sensors-20-03054-f014:**
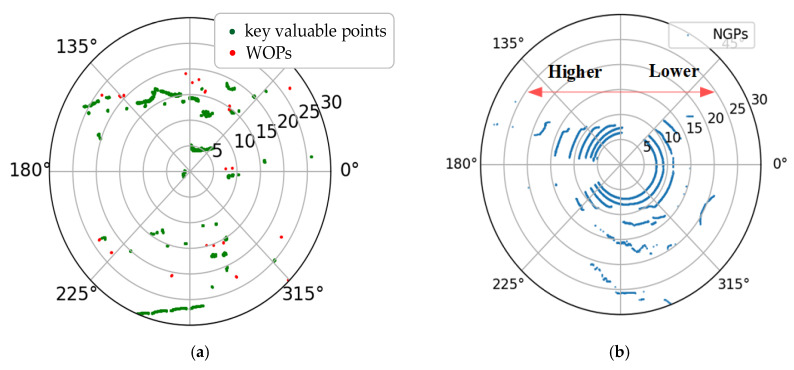
The key value points cluster in the key region with an inclination angle between the BGP and the projection plane. (**a**) The key valuable points in the key region. (**b**) The normal ground points in the key region.

**Figure 15 sensors-20-03054-f015:**
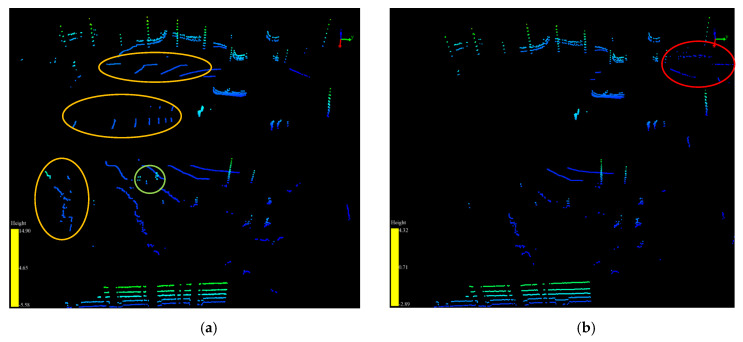
The AGPs and VOPs extraction results with and without fitting of the inclination angle between the BGP and the projection plane. (**a**) The extraction result with fitting of the inclination angle. (**b**) The extraction result without fitting of the inclination angle.

**Figure 16 sensors-20-03054-f016:**
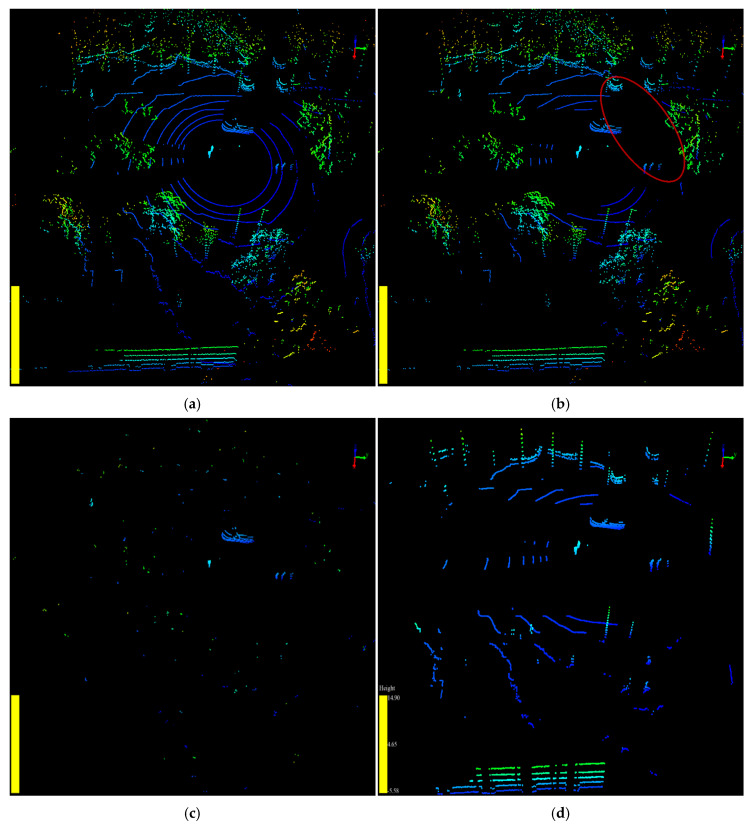
The point cloud before and after background point filtering by different algorithms are shown in VeloView. (**a**) The raw point cloud before background point filtering. (**b**)The point cloud after being filtered by random sample consensus algorithm (RANSAC); (**c**) The point cloud after being filtered by 3-D-DSF; (**d**) The point cloud after being filtered by slice-based projection filtering algorithm (SPF).

**Figure 17 sensors-20-03054-f017:**
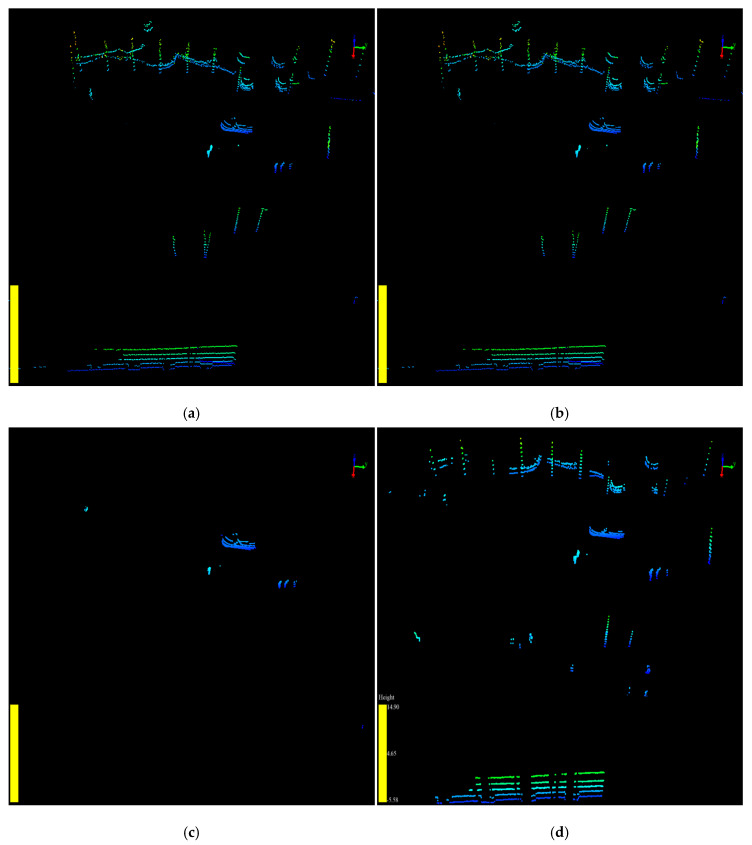
The VOPs before and after filtering are shown in VeloView. (**a**) The raw VOPs before filtering. (**b**)The VOPs after being filtered by RANSAC; (**c**) The VOPs after being filtered by 3D-DSF; (**d**) The VOPs after being filtered by SPF.

**Figure 18 sensors-20-03054-f018:**
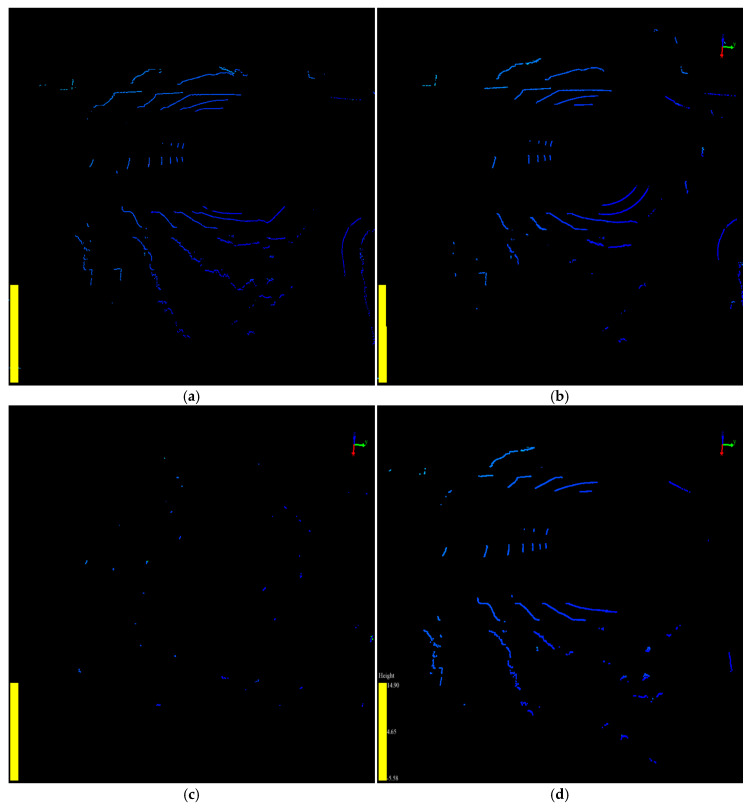
The AGPs before and after filtering are shown in VeloView. (**a**) The raw AGPs before filtering. (**b**)The AGPs after being filtered by RANSAC; (**c**) The AGPs after being filtered by 3D-DSF; (**d**) The AGPs after being filtered by SPF.

**Table 1 sensors-20-03054-t001:** The number of VOP extraction with or without fitting of the inclination angle between the BGP and the projection plane.

Fitting the Inclination Angle	The Key Region		Total
	Left Side	Right Side	
Without fitting	1939	1453	3392
With fitting	3054	1422	4476

**Table 2 sensors-20-03054-t002:** The background point filtering results by different algorithms.

Point Cloud	Algorithms	Before Filtering	After Filtering	Filtering Percentage (%)
raw point cloud	RANSAC	28960	24347	15.9
	3-D-DSF	28960	956	96.7
	SPF	28960	5738	80.1
VOPs	RANSAC	5596	4943	11.7
	3-D-DSF	5596	867	84.5
	SPF	5596	4476	20.0
AGPs	RANSAC	1482	1378	7.0
	3-D-DSF	1482	48	96.8
	SPF	1482	1262	14.8
VOPs + AGPs	RANSAC	24347	6321	74.0
	3-D-DSF	956	915	4.29
	SPF	5738	5738	0.00

**Table 3 sensors-20-03054-t003:** The runtime of the three algorithms.

Algorithm	Runtime (ms)	Frame Needed in the Algorithm
RANSAC	198	single-frame
3-D-DSF	113	multi-frames
SPF	51	single-frame

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
