# Peer review of "Background Point Filtering of Low-Channel Infrastructure-Based LiDAR Data Using a Slice-Based Projection Filtering Algorithm"

_sensors, 2020, doi:10.3390/s20113054_

Round 1

Reviewer 1 Report

This paper proposed a method for removing unnecessary areas in low channel lidars. The efficiency is improved by projecting 3d data into 2d coordinates to reduce dimensions. The point cloud was divided into VOP, WOP, AGP, and NGP to effectively remove unnecessary areas. Lastly, ANN was used to effectively filter the sloped ground.

However, the overall explanation is not clear. Particularly, the explanation of the figure and equation is weak. Also, the algorithm1 is described with pseudocode and long description, but it is very difficult to understand.

  1. Equations (1) to (3) need to be clearly explained. D and R are not clearly explained. It would be better to add figures to explain.
  2. The delta R in Figure 4 and the Rth in the paper seem to have the same meaning. However, it is not mentioned in the paper.
  3. The explanation for Figure 9 should be more clear.
  4. Section 4.3.2 is missing.
  5. In BP-ANN, how did you calculate Ground Truth(tan)?
  6. In the experiment, it would be good to show the results of each filter before and after.

Author Response

Dear Reviewer:

     We wish to express our very deep appreciation, and the appreciation of all of us, to your great efforts and suggestions for our manuscript. They are valuable and very helpful for revising and improving our paper, as well as the important guiding to our researches.

     A point-to-point response to your comments are in the attachment.

     Best Regards,

     Ciyun Lin

Reviewer 2 Report

Generally speaking, it is an interesting paper describing a new algorithm for transportation application. The paper can be much improved if the following issues are addressed:

1) Language used in the paper is a problem. There are a lot of typos and grammatical errors. For example, it should be "important" rather than "importation" in line 83. The paper should have been proofread by a professional English writer before submission.

2) What currency is used for the prices in lines 71 and 77?

3) The discussion session is not adequate. The authors should highlight the contribution of their work to the body of knowledge.

4) Although the authors outlined an agenda for further studies, they did not discuss the limitations of the current research.

Author Response

(The authors gave the same response as above.)

Round 2

Reviewer 1 Report

1. In BP-ANN, the input(fig.13(b) X)data seems a single point. It would be better to explain in detail the process of calculating the inclination with a single point and BP-ANN.

2. Comparative experimental with and without considering the ground inclination is required.

Author Response

Dear Reviewer:

Manuscript ID: sensors-783844

Title: “Background Points Filtering of Low-Channel Infrastructure-based LiDAR Data Using Slice-based Projection Filtering Algorithm”

We wish to express our very deep appreciation, and the appreciation of all of us, to your great efforts and suggestions for our manuscript. They are valuable and very helpful for revising and improving our paper.

The  point-to-point response to your comments are in the attachment. The modification marked in blue in revised version.

Once again, We appreciate for your warm work earnestly, and hope that the correction will meet with approval. Please feel free to contact us with any questions. We are looking forward to your reply.

Best Regards,

Ciyun Lin
